# Enhancing the Functional Properties of Tea Tree Oil: In Vitro Antimicrobial Activity and Microencapsulation Strategy

**DOI:** 10.3390/pharmaceutics15102489

**Published:** 2023-10-19

**Authors:** Franco Antonio Manzanelli, Soledad Ravetti, Sofia Gisella Brignone, Ariel Gustavo Garro, Sol Romina Martínez, Mariana Guadalupe Vallejo, Santiago Daniel Palma

**Affiliations:** 1Centro de Investigaciones y Transferencia de Villa María (CIT VM), Villa María 5900, Argentina; manzanellifa@unvm.edu.ar (F.A.M.); sravetti@unvm.edu.ar (S.R.); 2Instituto Académico Pedagógico de Ciencias Humanas, Universidad Nacional de Villa María, Villa María 5900, Argentina; arielgarro10@unvm.edu.ar; 3Unidad de Investigación y Desarrollo en Tecnología Farmacéutica (UNITEFA), CONICET, Córdoba 5000, Argentina; sofiagisellabrignone@unvm.edu.ar (S.G.B.); mariana.vallejo@unc.edu.ar (M.G.V.); 4Departamento de Ciencias Farmacéuticas, Facultad de Ciencias Químicas, Universidad Nacional de Córdoba, Córdoba 5000, Argentina; 5Ministerio de Ciencia y Tecnología, Gobierno de Córdoba, Córdoba 5004, Argentina; 6Instituto de Investigación en Tecnologías Energéticas y Materiales Avanzados (IITEMA), CONICET, Departamento de Química, Facultad de Ciencias Exactas, Físico-Químicas y Naturales, Universidad Nacional de Río Cuarto, Río Cuarto 5804, Argentina; martinezsolr@gmail.com

**Keywords:** essential oil, biological activity, antimicrobial resistance, microcapsules, periocular infections

## Abstract

In the context of addressing antimicrobial drug resistance in periocular infections, Tea Tree Oil (TTO) has emerged as a promising therapeutic option. This study aimed to assess the efficacy of TTO against bacterial strains isolated from ocular infections, with a particular focus on its ability to inhibit biofilm formation. Additionally, we designed and analyzed microcapsules containing TTO to overcome certain unfavorable physicochemical properties and enhance its inherent biological attributes. The quality of TTO was confirmed through rigorous analysis using GC-MS and UV-Vis techniques. Our agar diffusion assay demonstrated the effectiveness of Tea Tree Oil (TTO) against ocular bacterial strains, including *Corynebacterium* spp., coagulase-negative Staphylococcus spp., and *Staphylococcus aureus*, as well as a reference strain of *Staphylococcus aureus* (ATCC 25923). Notably, the minimum inhibitory concentration (MIC) and minimum bactericidal concentration (MBC) for all tested microorganisms were found to be 0.2% and 0.4%, respectively, with the exception of *Corynebacterium* spp., which exhibited resistance to TTO. Furthermore, TTO exhibited a substantial reduction in biofilm biomass, ranging from 30% to 70%, as determined by the MTT method. Through the spray-drying technique, we successfully prepared two TTO-containing formulations with high encapsulation yields (80–85%), microencapsulation efficiency (90–95%), and embedding rates (approximately 40%). These formulations yielded microcapsules with diameters of 6–12 μm, as determined by laser scattering particle size distribution analysis, and exhibited regular, spherical morphologies under scanning electron microscopy. Importantly, UV-Vis analysis post-encapsulation confirmed the presence of TTO within the capsules, with preserved antioxidant and antimicrobial activities. In summary, our findings underscore the substantial therapeutic potential of TTO and its microcapsules for treating ocular infections.

## 1. Introduction

The pharmaceutical industry continually explores novel therapeutic alternatives for preventing and treating various diseases. Focusing on those that come from natural sources, an attractive option are essential oils (EOs) biosynthesized by plants. These natural products are mainly obtained by distillation, applying different conventional and non-conventional extraction techniques [1]. This is feasible because EOs are a mixture of volatile substances of diverse chemical composition, mainly terpenes, phenylpropanoids, and aromatic derivatives, which vary according to the species and Ambiental and anthropogenic factors.

EOs have been used for thousands of years in primary health care, demonstrating a broad spectrum of pharmacological activities [2,3]. Today, it is known that these medicinal properties are given because the components of EOs are capable of modulating numerous multiple signal transduction pathways, individually or in a synergistic manner.

Among the diverse pharmacological activities, antimicrobial properties of EOs have gained importance, particularly in light of the escalating challenge of microbial resistance [4]. In this scenario, EOs are surging as potential alternatives to antibiotics or as a complementary therapy alongside them.

One often overlooked healthcare issue pertains to periocular infections, which affect the area around the eyes, including the eyelids and the surrounding region. These infections can be caused by bacteria, viruses, or fungi. When left untreated or inadequately managed, they can progress and directly impact the eyes, leading to conditions like conjunctivitis, orbital cellulitis, chronic blepharitis, chorioretinitis, and endophthalmitis [5].

Both periocular and ocular infections are commonly treated with broad-spectrum antimicrobial drugs, often without proper pathogen identification through culture and susceptibility testing. This misuse can potentially promote antimicrobial resistance in ocular bacteria. In particular, bacterial biofilms are key contributors to resistance mechanisms, protecting the bacterial community [6].

Limited studies have been conducted on the use of essential oils (EOs) against bacteria from ocular infections, and even fewer studies on encapsulation systems that include EOs for this kind of pathology are reported. However, using EOs in pharmaceutical applications presents challenges due to their physicochemical properties. Their unfavorable physicochemical properties, including hydrophobicity, low solubility in aqueous media, high volatility, oxygen-mediated decomposition, and an undesired biological profile, like significant irritant action, restrict their applicability as therapeutic agents. Cosmetic and pharmaceutical formulations incorporating EOs have been developed to address these issues, yet stability problems persist, due to exposure to environmental factors like air, heat, light, and moisture substantially altering their composition during storage. Encapsulation technologies, particularly microencapsulation via spray drying, offer a promising solution [7,8,9,10].

Microencapsulation is an increasingly favored technique in the pharmaceutical industry due to its flexibility, cost-effectiveness, and suitability for heat-sensitive compounds [7,8,9,10]. It enables the production of ultrafine solid structures with high stability and encapsulation efficiency.

Incorporating antibacterial EOs into pharmaceutical formulations for periocular infections has the potential to enhance treatment efficacy, reduce the risk of antimicrobial resistance development, prevent its spread to ocular structures, and minimize the risk of serious complications that could endanger vision. Moreover, the complementary pharmacological properties of EOs, such as its antioxidant and anti-inflammatory effects, have the potential to enhance overall ocular health. These properties fortify the immune system and shield ocular tissues from oxidative damage caused by inflammatory processes.

Tea Tree Oil (TTO) presents a promising profile for antimicrobial therapy. This oil is obtained mainly by steam distillation of the leaves of *Melaleuca alternifolia* (Cheel) Myrtaceae, a tree native to Australia [11,12]. ISO 4730:2017 standards establish that the main component of TTO is terpinen-4-ol, in a proportion not less than 40% [13]. Different studies on the subject have demonstrated the broad-spectrum antimicrobial activity of TTO, including antibacterial, antiprotozoal, antifungal, and antiviral activity [14,15,16,17].

With the passage of time, the gradual oxidation of components within TTO during the storage period can lead to a decrease in its antimicrobial effectiveness and potentially initiate undesired chemical reactions. Consequently, there is a growing demand for formulations that not only preserve the integrity of TTO but also enhance its inherent biological properties.

In view of these considerations, this study pursued a dual objective. Firstly, it assessed the antibacterial activity of a natural extract, TTO, against bacterial strains isolated from ocular infections. Secondly, it developed an encapsulation methodology using the spray drying technique to microencapsulate the selected EO. The resulting microcapsules underwent various analyses, encompassing the evaluation of their physical and morphological characteristics, in vitro drug release profiles, and investigations through scanning electron microscopy.

Therefore, further research and exploration of TTO as a promising therapeutic option for eye infections is essential.

## 2. Materials and Methods

### 2.1. Materials

TTO was purchased from Thursday Plantation^®^ (Ballina, Australia). The concentration of TTO is 1 mL per milliliter (1 mL/mL). Terpinen-4-ol and cyclohexanol were obtained from Sigma Aldrich (Buenos Aires, Argentina). Arabic Gum (AG), maltodextrin (MDX), and silicon dioxide (SD) were purchased from Pura Química (Córdoba, Argentina) and were used as wall materials to formulate the microparticles.

2,2-diphenyl-1-picrylhydrazyl (DPPH) and 3-(4,5-dimethylthiazol-2-yl)-2,5-diphenyl tetrazolium bromide) (MTT) were obtained from Sigma-Aldrich, Merck (Buenos Aires, Argentina). Solvents of analytical reagent grade (tween80, glycerol, ethanol, and dimethyl sulfoxide (DMSO) ~95.0%) and solid reagents for buffers were acquired from Anedra (Buenos Aires, Argentina). Crystal violet (CV), Mannitol salt agar (MSA), Muller Hinton agar (MHA), Muller Hinton broth (MHB), and Tryptic soy broth (TSB) were purchased from Britania (Buenos Aires, Argentina). Clinical bacterial isolates and reference strains (American Type Collection Culture, ATCC) were supplied at the microbiology laboratory from Hospital Regional Pasteur, Villa María, Córdoba, Argentina. Villa Maria belongs to the Latin American Antimicrobial Resistance Surveillance Network (Whonet-Argentina).

### 2.2. TTO Analysis

#### 2.2.1. Gas Chromatography—Mass Spectrometry Analysis

TTO verified the specifications given by the ISO standards referred to its components by gas chromatography coupled to mass spectrometry (GC-MS). Qualitative and quantitative analysis of TTO were performed using Clarus SQ8 equipment (Perkin Elmer, Walthman, MA, USA) with an Agilent DB-5 column, 30 m in length, 0.25 mm of diameter, and 0.25 µm thickness of the film. A temperature program was adapted from the method reported by Tranchida et al. [18]. Initial temperature was set to 50 °C, increasing 3 °C/min until 150 °C, and remained until complete a total time of 35 min. Injector temperature and detector temperature were 280 °C. Solvent delay was 4 min, helium was employed as carrier gas at 1 mL/min, and split injection mode was selected. Spectrums were acquired in a single quadrupole mass spectrometer, under vacuum, with an ionization energy of −70 eV. Mass range was set to 51–400 Da. TurboMass 6.1.0. software was used to acquire and process data. Fragmentation patterns of the obtained signals were compared with those included in the NIST mass spectral library and, therefore, identified as regular components of TTO [13]. The percentage of terpinen-4-ol was established by calculating the rate of the individual area and total area.

#### 2.2.2. Characterization and Validation of UV/Vis Method

The TTO was analyzed using UV/Vis spectroscopy using a spectrophotometer (Analytik Jena, Specord S600, Jena, Germany) with a sample to determine its complete absorption spectrum in absolute ethanol. Variable absorbance spectroscopy scans the entire UV and visible wavelength range to obtain detailed information about the sample’s absorption. The highest absorbance was obtained at 265 nm, coinciding with the value reported in the literature [19]. Therefore, the total content of TTO components was established by UV/Vis. A calibration curve was built using six dilutions of TTO in absolute ethanol (three replicates each) over a range of 3.58 µg/mL to 89.5 µg/mL. The analytical procedure was validated according to the following criteria: linearity was established from a calibration curve applying least-square linear regression analysis and correlation coefficient (r); accuracy and precision were evaluated by processing replicates of samples (n = 6) expressing the results as relative standard deviation (RSD). The absorbance of TTO solutions at λmax = 265 nm was measured. In addition, a standard equation y = 0.0106x − 0.0008 (R^2^ =  0.9995) was obtained (where “y” represents the absorbance, and “x” represents the oil concentration (µg/mL)).

### 2.3. Biological Activity of Free TTO

#### 2.3.1. Antimicrobial Screening

##### Bacterial Strains

The present study was conducted using three clinical isolates obtained from eye infections, in addition to the reference ATCC strain: *Staphylococcus aureus* (ATCC 25923). The clinical strains were isolated from the conjunctiva (*Corynebacterium* spp. and *Staphylococcus* spp., negative for coagulase) and the cornea (*Staphylococcus aureus*).

Cultures were stored in 10% glycerol (*v*/*v*) at −80 °C. Bacterial strains were grown aerobically in MHA at 37 °C for 18 h; subsequently, the bacterial culture was prepared by inoculating one single isolated colony from a pure culture in MHA.

##### Diffusion Agar Assay

The agar diffusion method was employed to assess the antimicrobial effectiveness of TTO. A microbial suspension of 0.5 McFarland was prepared for each microorganism and then introduced into MHA plates using a sterile cotton swab. In total, 25 µL of free TTO at different concentrations (895 mg/mL to 56 mg/mL) were tested in triplicate. The inhibition zone produced by each microorganism was observed after 24 h and measured using a Vernier caliper.

##### Determination of the Minimum Inhibitory Concentration (MIC) and Minimum Bactericidal Concentration (MBC)

The antimicrobial activity of TTO was initially assessed using the Clinical and Laboratory Standards Institute (CLSI) protocol for antimicrobial susceptibility by agar diffusion [20].

The MIC was defined as the lowest TTO concentration without visible growth, and the MBC was defined as the lowest concentration reducing the initial inoculum by ≥99.9%.

In the subsequent phase, MIC and MBC were determined using the microdilution method. The isolated microorganisms were cultured on MHA at 37 °C until they reached the exponential growth phase. Serial dilutions of TTO were prepared in a 96-well plate, whereas flat-bottom 96-well microplates were filled with 100 µL of MHB per well, except for the first row, which contained 200 µL of TTO.

To overcome the insolubility of TTO in the medium, it was supplemented with Tween 80 detergent at the final concentration of 0.05%. A diluted bacterial suspension was added to each well to achieve a concentration of ~10^6^ colony-forming units (CFU/mL). Controls were run in parallel.

##### Bacterial Kill Curves

Time kill curves illustrated bacterial elimination over time as a function of the concentration of TTO. The CLSI protocol was followed. The determination is made by counting the number of viable cells at different times by subjecting the bacterial inoculum to TTO. The bactericidal profile was established when there was a decrease of 3 logarithmic units in a given time. The bacterial death curves were made taking the results obtained in the determination of the MIC as a starting point.

Bacterial cultures in the exponential growth phase were suspended in MHB until reaching a concentration of approximately ~10^8^ CFU/mL. Next, the inoculum was adjusted to a concentration of ~10^6^ CFU/mL to be inoculated against the different TTO dilutions. Bacterial viability counts were performed at 0, 1, 2, 3, and 4 h of incubation at 37 °C for 24 h. Different aliquots of each sample were collected, and serial dilutions were run [21,22,23].

The lethality curves were represented graphically expressing the log10 CFU/mL as a function of time.

#### 2.3.2. Evaluation of Antibiofilm Activity

##### Crystal Violet Assay

Starting from fresh cultures, a dilution was made in TSB, adjusting the inoculum concentration to ~10^6^ CFU/mL. To induce biofilm formation, 200 μL of this suspension was added to each well of a sterile polystyrene plate and incubated for 24 h at 35 °C with continuous agitation (130 rpm). Subsequently, once the biofilm was formed, the supernatant was discarded, and the wells were washed three times with 200 μL of PBS buffer to remove planktonic cells. Solutions were prepared in TSB at different *% v*/*v* concentrations (0.8, 1.5, 12.5, and 25). After the final wash, 200 μL of these solutions was added, and the plate was incubated for 24 h at 35 °C in a shaker with continuous agitation at 130 rpm. Finally, the biomass of the biofilm was analyzed in triplicate using the absorbance spectrophotometry method on a microplate reader (Thermo Scientific—Multiskan FC, Munich, Germany). To quantify the biofilm biomass, Crystal violet (CV) staining assay was employed through a microplate reader (Thermo Scientific—Multiskan FC). The percentage reduction in biofilm biomass was calculated with respect to the average OD570 obtained from the wells that were not incubated with TTO.23 Biofilm biomass reduction (%) was calculated based on the relevant control by applying the following equation: reduction % = [(Abs0 − Absx)/Abs0] × 100, where Abs0 ~570 is the absorbance of the controls to which no treatment with TTO was added, and Absx ~570 absorption corresponds to the absorbance of the sample after applying the treatment with TTO [22,23].

##### MTT in Assembled Biofilms

Bacterial viability was evaluated using tetrazolium salts, where it is reduced to formazan in the presence of live cells, and its absorbance can be measured at 570 nm. The exposure of biofilms to TTO treatment and controls was performed as described above. Subsequently, the samples were washed with 200 μL of MTT reagent (200 μg/mL in PBS) added to each well and incubated in darkness conditions for 3 h at 35 °C. Then, the supernatant was discarded, 150 μL of DMSO was added to solubilize crystal formation, and the absorbance was recorded in a microplate reader (Thermo Scientific—Multiskan FC). Viability (%) was calculated as described above [23].

### 2.4. Antioxidant Activity of TTO

The antioxidant potential of both the free and microencapsulated TTO was measured using the DPPH method. The DPPH free radical scavenging potential was determined according to the method of Brand-Williams et al. [24], with equal quantities of TTO and microcapsules. The antioxidant activity was determined by preparing 5 dilutions (1, 10, 100, 500, and 1000 μg/mL). As control, various solutions of ascorbic acid were employed due to its well-known antioxidant capacity. The analysis was performed in a microplate by adding 2 mL of the solution and 2 mL DPPH methanolic solution. After 30 min of incubation at room temperature in the dark, the absorbance was measured at 517 nm using a spectrophotometer (Analytik Jena, Specord S600, Jena, Germany) with a microplate reader. 

### 2.5. Preparation of Microcapsules by Spray-Drying

#### 2.5.1. Preparation of Emulsions

The emulsions (1 and 2) were prepared using MDX and AG as carrier (well material). MDX and AG were previously dissolved in distilled water at 50 °C for 1 h and left to stand for 24 h at room temperature. The next day, SD was added as a lubricant. SD, MDX, and AG were used in a proportion of 2:1:1, respectively. For the emulsion preparations, TTO was incorporated into the wall material emulsion using a high-power homogenizer (Proscientific PRO 250, Oxford, UK) at 24,000 rpm for 5 min. Immediately after the emulsification process, the TTO emulsion was dried by spray drying. The schematic representation of the microencapsulation formation process is depicted in Figure 1.

#### 2.5.2. Spray Drying

Spray drying was performed using a laboratory-scale Mini Spray Dryer (Büchi B-290, Büchi Labortechnik AG, Flawil, Switzerland). The samples were atomized with a hot air stream in the drying chamber. A two-fluid nozzle of 0.5 mm cap orifice diameter was used. This type of nozzle operates on the basic principle of utilizing high-speed air to crush the liquid, resulting in smaller liquid particles and higher flow rates. The following parameters were fixed. For emulsion 1: pump 10, aspirator 100; Q-flow, 600 L/h; inlet temperature, 130 °C; and outlet temperature, 100 °C. The same parameters were used for emulsion 2, except for the inlet temperature, which was 120 °C, and the pump was 7, respectively.

#### 2.5.3. Determination of Microencapsulation Yield (EY), Microencapsulation Efficiency (ME) and Oil Embedding Rate (ER)

The EY (%) was calculated as the ratio between the recovered solids (g) after the spray drying process and the initial solids of the formulation (g) using Equation (1).

The theoretical oil content was determined using Equation (2), where Moil, MMX, MAG, and MSD are the masses (g) of TTO, MX, AG, and SD added in the system, respectively.

The ME (%) was calculated as the ratio of the total oil content obtained inside the microcapsules and the surface oil content (Equation (3)). To measure the total oil content, a sample of microcapsules (100 mg) was put into 15 mL anhydrous ethanol. After sonication for 1 h, the microcapsules were filtered and washed with 10 mL and then another 5 mL of ethanol. All the filtrate was put together. Moreover, to measure the surface oil content, 100 mg of microcapsules was put in a funnel and washed with 5 mL ethanol 3 times. Both filtrates were put together to measure the oil content. The absorbance was measured with a UV/Vis spectrophotometer, and the total oil content in the sample was calculated according to the standard equation. 

The ER oil embedding rate was defined according to Equation (4). M2 was the total oil mass obtained in 100 mg microcapsules, and M1 was the oil mass obtained on the surface of 100 mg microcapsules.
(1)$$EY=\frac{recovered\;solids\;(g)}{initial\;solids\;(g)}\times 100\%$$
(2)$$Theoretical\;oil\;content\;(\%)=\frac{Moil}{Moil+MMX+MGA+MSD}\times 100\%$$
(3)$$ME=\frac{M2-M1}{M2}\times 100\%$$
(4)$$ER=\frac{Total\;oil\;content}{theoretical\;oil\;content}\times 100\%$$

#### 2.5.4. Particle-Size Analyses

The particle size distribution was measured using a laser scattering particle size distribution analyzer (Horiba Partica LA-960, Kyoto, Japan).

The span calculation is the most common format to express distribution width.
(5)$$Span=\left[(d90-d10)/d50\right]$$

The value of the span was calculated applying the following equation (Equation (5)): where d10, d50, and d90 correspond to the diameters at 10, 50, and 90% of the cumulative particle size distribution.

#### 2.5.5. Scanning Electron Microscopy (SEM)

Morphology and surface features of spray-dried microcapsules were evaluated using scanning electron microscopy (SEM) (ZEISS, Ʃigma, Oberkochen, Germany) operated at 5 kV with magnifications of 5000× in Lamarx Laboratories (Universidad Nacional de Córdoba, Argentina). Previously, the samples were attached to a double-sided adhesive tape mounted on SEM stubs and metallized with gold/palladium under vacuum. 

#### 2.5.6. Fourier-Transform Infrared Spectroscopy (FTIR)

Fourier-Transform Infrared Spectroscopy (FTIR) analysis of TTO in its pure form, as well as the mixture of wall material and structures after spray drying, was performed on a droplet of each sample. The equipment used was a CARY 630 FTIR (Agilent Technology, Santa Clara, CA, USA), covering a range of 500 to 4000 cm^−1^ with a resolution of 3 cm^−1^. Sixteen scans were performed for each sample analyzed.

#### 2.5.7. Antimicrobial Screening of Microencapsulated TTO

The antibacterial effect of the microparticles was evaluated by completely releasing the encapsulated TTO against the previously mentioned strains *S. aureus* and *S. aureus* ATCC. For the assay, Formulation 1 was selected, and tubes were prepared with samples corresponding to 50, 100, 200, and 300 mg of microcapsules with 0.5 mL of DMSO and 0.5 mL of MHB. Subsequently, the sample was sonicated for 1 h and then centrifuged at 10,000 RPM at 5 °C for 10 min (ThermoST16R). These samples were inoculated with 1 mL of bacterial suspension to achieve a concentration of approximately ~10^6^ CFU/mL for the purpose of evaluating the MIC and MBC. Parallel controls were performed in MSA.

## 3. Results and Discussion

### 3.1. GC-MS Analysis of TTO and Encapsulated TTO

The composition of the commercial sample of TTO used was compared with that reported in the literature. The percentage of the chemical marker stated by ISO 4730:2017 [13], terpinen-4-ol, was determined as 44.8%, as well as other highlighted components (Figure 2A). It is important to point out that o-cymene, a decomposition indicator, was detected at a low percentage in this analysis.

On the other hand, after microencapsulation, the composition of TTO inside the capsule (Formulation 1) was also analyzed, and a similar fingerprint was observed, where terpinen-4-ol was the main component (52.1%) (Figure 2B).

### 3.2. Biological and Antioxidant Activity of Free TTO

#### 3.2.1. Antimicrobial Activity of TTO

The antimicrobial activity of TTO was evaluated at different concentrations. The inhibition zone produced by each microorganism (Table 1) was measured using a Vernier caliper.

The agar diffusion method was applied to test the antimicrobial properties of TTO, and it showed activity against studied microorganisms. It was observed that TTO inhibited cellular growth. As the concentration of TTO increases, the inhibition zone (halo) of microbial growth also enhances. The agar diffusion method is commonly used as a quick test to determine the potential susceptibility or resistance to an antibiotic. The results can be compared with those of Mumu (2018) [25], where inhibition zones were observed by TTO against clinical bacterial isolates after 24 h of incubation. Although the study shows that TTO is effective against both Gram-positive and Gram-negative strains, our study found inhibition values higher than those found in this study for *S. aureus* isolates. However, for new molecules, there are no standard reference measurements for comparisons, thus the agar diffusion method becomes a qualitative tool for checking the inhibition produced by these active molecules, and it should only be used as a screening test [26].

#### 3.2.2. MIC/MBC

After using the agar diffusion method as a screening, the antimicrobial activity was evaluated more accurately through determinations of MIC and MBC using a broth dilution technique. The values are shown in Table 2.

The lowest dilution for MIC in the tested microorganisms was 0.2% *v*/*v* (1.8 mg/mL) of TTO, whereas it was 0.4% *v*/*v* for MBC (3.5 mg/mL).

Both the clinical and collection strains of *S. aureus* exhibited the same MIC and MBC values, whereas higher concentrations were required for *Staphylococcus* spp. negative coagulase to achieve inhibitory and bactericidal effects. However, these values align with reports from other authors [27,28]. Moreover, it has been described that *S. aureus* requires concentrations between 0.5–1.25% *v*/*v* (4.5–11.2 mg/mL) for MIC and 1–2% *v*/*v* (9.0–18.0 mg/mL) for MBC, whereas, in this study, the same effect was found with concentrations of 0.2% *v*/*v* (1.8 mg/mL) and 0.4% *v*/*v* (3.5 mg/mL), respectively. As for the *Corynebacterium* spp. isolate, no MBC was found, whereas the MIC was found at a concentration of 0.4% (3.5 mg/mL).

According to the literature, it has been reported that bacteria are susceptible to TTO at concentrations of 1.0% (9.0 mg/mL) or less. However, higher MIC values have been disclosed for other Gram-positive isolates, such as *Staphylococcus* and *Micrococcus*, and Gram-negative isolates, such as Pseudomonas aeruginosa [29,30].

TTO is predominantly bactericidal in nature, although it can be bacteriostatic at lower concentrations [12].

The studies on essential oils as antimicrobial agents have focused on their abilities to kill different types of microorganisms, including bacteria, fungi, and viruses [31]. Results show that many essential oils have potent antimicrobial activities and can be effective against a variety of pathogens, including some bacterial strains resistant to conventional antibiotics. Moreover, for an EO to be considered an active compound, the extract must have an approximate MIC of less than 1 mg/mL [32].

### 3.3. Bacterial Kill Curves of Free TTO

Bacterial death curves are a method employed to determine the in vitro activities of different concentrations of a compound against a microorganism over a specific period. The American Society for Microbiology established that any new approach claiming to be antimicrobial or antibacterial must achieve a reduction of at least 3 logarithms in CFU [33]. This reduction indicates that the antimicrobial agent has been effective in eliminating approximately 99.9% or more of the bacterial culture, demonstrating its bactericidal activity.

Various concentrations of TTO were utilized to assess its efficacy over time as an antimicrobial agent. The selected strains were isolated from ocular infections, as previously mentioned (Section 2.3.1), and a reference strain was also included. Figure 3 presents the outcomes of these experiments along with their respective controls. In the case of *Staphylococcus* coagulase-negative, bacterial death was observed with 0.2% TTO after 1 h. Conversely, for *S. aureus* ATCC and *S. aureus*, the same effect was observed with the same concentration of TTO after 2 h. However, no bacterial death was observed for *Corynebacterium* spp.

Li et al. 2016 noted a trend in which, with increasing concentrations of TTO, the rate of cell death and the duration of the growth lag phase correspondingly increased. These findings indicated that TTO exhibited time- and concentration-dependent antibacterial effects [12,34].

The mechanism of action of TTO is primarily attributed to its monoterpenoid components, which are the major bioactive constituents responsible for its antimicrobial properties. Monoterpenoids are a class of naturally occurring organic compounds with a characteristic molecular structure. They have demonstrated potent antimicrobial activities against a wide range of microorganisms, including bacteria, fungi, and viruses.

### 3.4. Antibiofilm Activity of Free TTO

Biofilm is considered a virulence factor that promotes the survival, resistance, and pathogenic capacity of microorganisms, complicating treatments for infections and significantly increasing the severity of diseases. With the purpose of exploring the effectiveness of TTO, we decided to test the antibiofilm effect. Biofilm formation was induced in all ocular bacterial strains. Biofilm generation after 24 h was quantified by CV assay, indicating that all stains yielded positive biofilm formation, ~OD570 > 0.24 [35]. The performance of TTO against mature biofilms was evaluated using CV and MTT assays, the former yielding information regarding the overall biomass quantification, whereas the latter evaluates bacterial viability, proliferation, and cellular cytotoxicity. Figure 4 displays the results of these experiments.

In all strains, a reduction in biofilm biomass between 30 and 70% was observed when treated with TTO. No differences were observed in the reduction in biomass between the different concentrations when CV assays were performed. However, when cell viability was evaluated using MTT, greater effectiveness was observed when the applied treatments were at lower concentrations of TTO. A reduction of over 80% was achieved in treatments with 1.6% *v*/*v* TTO in 3 (A, B, and D) out of the four strains used. On the other hand, C required a more diluted concentration (0.8% *v*/*v*) to achieve greater effectiveness compared to the rest of the treatments. There are some precedents, considering that the minimum eradication concentration of mature biofilms formed by *S. aureus* bacteria was two times higher than the MIC but never higher than 1% of TTO [36]. Furthermore, the use of essential oils at low concentrations has often been effective in inhibiting biofilm formation in pathogenic strains [37].

### 3.5. Antioxidant Activity

Before encapsulation, TTO showed an antiradical activity above 80%. After the encapsulation process, the percentage was 60%.

The antioxidant properties of natural compounds can help protect cells from damage caused by free radicals, which are unstable molecules that can harm cells and contribute to aging and disease development. In this study, different concentrations of TTO were tested, and ascorbic acid was used as a control, known for its antioxidant property. A 40% activity was observed at a concentration of 1 µg/mL of TTO, whereas a 90% activity was achieved at a concentration of 1 mg/mL in a similar study conducted by Kim et al., 2004, in a methanolic solution, yielding comparable results, with methanol exhibiting approximately 80% free radical scavenging activity [38].

Regarding the antioxidant capacity of microencapsulated TTO, the tested activity was maintained in the same proportion.

### 3.6. Microencapsulation of TTO

#### 3.6.1. Microencapsulation Yield (EY), Microencapsulation Efficiency (ME), and Oil Embedding Rate (ER)

The percentage of microparticle recovery (% microencapsulation yield) is the amount of powder obtained in the spray dryer collector relative to the total amount of solid content prior to the process. The EY after the spray-drying procedure was estimated at 80–85%.

The increase in the initial solid quantity could also lead to an increase in powder yield. However, a study involving the microencapsulation of lavender essential oil showed that a high concentration of solids can decrease the emulsion’s water content, reducing the time needed to form a membrane on the particle surface during spray drying. This resulted in an efficient powder recovery [39].

The ME was about 90–95% for both microcapsule preparations. The ME of TTO is the ratio between the total TTO within the microcapsules and the surface TTO. The ER, calculated as the ratio between the total TTO and the theoretical TTO, was about 40%. This value indicated the loading capacity and the degree of success of the coating in preventing the negative effects of essential oils, such as volatilization. For medicinal applications, it is not practical to use a formulation with low quantities of active pharmaceutical ingredients, even if they provide high encapsulation efficiencies, as a significant amount of polymer would be required to achieve the necessary therapeutic dose [39].

#### 3.6.2. Particle Size and Morphological Characterization of the Microcapsules of TTO

The samples were atomized with a hot air stream in a drying chamber, thus making it possible to obtain solid microparticles when the EO was trapped within a film of encapsulating material.

Particle size is important, as it can affect the microencapsulation efficiency, as well as the interaction with other fluids. The particle size of a microcapsule ranges from 1 to 1000 µm depending on the manufacturing method and its specific purpose. Particle size is considered a critical aspect for encapsulating substances and is an important factor in the controlled release of bioactive agents [40,41].

All the parameters described in the methods were analyzed (Table 3), resulting in an average diameter (d50) of 6 µm for both formulations. The most relevant value was that of d90, which characterizes the majority of the particle population. The D90 values obtained were 12.90 ± 0.30 µm and 12.20 ± 0.30 µm for Formulations 1 and 2, respectively, reflecting a uniform particle size, as expected for them to be considered microparticles. Furthermore, the calculated span value demonstrates a narrow particle size distribution for both formulations across all replicates.

A slightly broader particle distribution was recorded, ranging between 8 and 15 μm, when microencapsulating the EO of *Lippia sidoides Cham. (Verbenaceae)* [42] and the microencapsulation of *Origanum vulgare* L. essential oil (7–18 μm) using a combination of MX, AG, and modified starch [43].

In our study, the smallest particle size was recorded with Formulation 2 (12.20 μm). However, no significant differences were found in particle size compared to Formulation 1 when variations in equipment parameters were made prior to spray drying. For these reasons, both formulations could be used as potential strategies for microencapsulating TTO.

With the purpose of ensuring proper administration of a microencapsulated active ingredient, it is advisable for the microparticle size to be sufficiently small [44].

The obtained SEM images play a crucial role, not only in verifying the acquired particle size data but also in assessing the morphological features that elucidate the functional attributes of the particles, such as their capacity to retain and safeguard the bioactive ingredient [45].

The images captured revealed distinct individual spherical structures with smooth surfaces, devoid of any cracks or fissures that might allow the release of TTO. This ensures the sustained preservation of its functionality over time.

The SEM images are consistent with the obtained span value. While a range of sizes is evident, which is a characteristic feature of particles produced by spray drying due to factors like the material used, formulation, and atomization process [41], the observed size range remains relatively narrow. In a study involving the microencapsulation of lavender essential oil, it was observed that smaller particles fused with larger ones, as evident in images B and D [39,46].

The particle sizes extracted from the SEM images were manually measured by identifying the largest corresponding dimension. The results indicated a particle diameter ranging from 3 to 19 µm (Figure 5C–F), with the majority falling within the intermediate range. This reaffirms the presence of a narrow particle size distribution. Considering that the hydrodynamic diameter, determined via DLS, is derived from analyzing the sample in a liquid state (without affecting the sample’s aggregation state), whereas the diameter measured for the particles through SEM necessitates the preparation of the sample on a support, such as a thin film, followed by drying with additional carbon deposition, the consistency between the results is highly commendable. This further underscores that the particles are accurately represented by the solid sphere model.

The FTIR spectra of TTO and the microcapsules with and without TTO are presented in Figure 6.

The spectrum of TTO revealed a stretching vibration peak corresponding to the C-H bond at 2960 cm^−1^. Additionally, it exhibited numerous peaks within the wavelength range of 1550 cm^−1^ to 600 cm^−1^. The peak at 1126 cm^−1^ was attributed to the stretching vibration absorption peak of the C-O bond in the tertiary alcohol of terpenes and terpineol, respectively. The peak at 924 cm^−1^ corresponded to the bending vibration absorption peak of an unsaturated double bond.

For the microcapsules without TTO, characteristic hydroxyl peaks (O-H stretching) were observed around 3332 cm^−1^. Peaks corresponding to the C-H stretching from the carboxylic group appeared around 2929 cm^−1^. Peaks corresponding to amine or carbonyl groups appeared at 1608 cm^−1^. For the microcapsules with TTO, characteristic hydroxyl peaks (O-H stretching) were observed around 3298 cm^−1^. Peaks corresponding to the C-H stretching from the carboxylic group appeared around 2930 cm^−1^. Peaks corresponding to amine or carbonyl groups appeared at 1603 cm^−1^. All of these peaks were characteristic of the wall material used in the formulation design. These bands representing coating materials were evident in the microcapsule spectra. This implies that these carbohydrates maintained their structures during the drying process.

The spectra of the microcapsules with and without TTO confirmed the successful microencapsulation through spray drying, with no TTO present on the surfaces of the microcapsules [47,48].

### 3.7. Antimicrobial Screening of Microencapsulated TTO

A clinical strain and a reference strain of *S. aureus* were employed to evaluate the antimicrobial effectiveness of TTO microcapsules. Bacterial samples were exposed to different concentrations of microcapsules (50, 100, 200, 300, and 400 mg) for 24 h.

Afterward, cell suspensions were 10-fold diluted and plated on MHA, and, in parallel as control, bacteria were harvested in a selective and differential media such as MSA to encourage the proper growth of *S. aureus*. Regarding the culture media, growth inhibition was observed, starting at 100 mg of microcapsules for *S. aureus* ATCC and 200 mg for *S. aureus*. Various concentrations of Formulation 1 were meticulously employed to assess its antimicrobial effectiveness over a period spanning 24 h. The chosen test strains for this evaluation included a reference strain of *S. aureus* ATCC and a clinical isolate of *S. aureus*. The graphical representation in Figure 7 eloquently illustrates the profound impact of Formulation 1 on cellular viability, showcased over the course of the incubation period. The data presents a strikingly evident bactericidal profile for both of the aforementioned bacterial strains, underscoring a significant reduction in bacterial viability, surpassing an impressive 99.99% reduction concerning the control group. These results serve as compelling evidence of the successful and efficacious release of the active compound, TTO, within Formulation 1, solidifying its potential as a potent antimicrobial agent.

## 4. Conclusions

TTO has demonstrated efficacy against bacterial strains isolated from ocular infections, including *Corynebacterium* spp., *Staphylococcus* spp. negative for coagulase and *Staphylococcus aureus*, as well as a reference strain of *Staphylococcus aureus* (ATCC 25923). TTO exhibited a substantial reduction in biofilm biomass, ranging from 30% to 70%.

The microencapsulation technique used in the study successfully prepared TTO-containing formulations with high encapsulation yields, microencapsulation efficiency, and embedding rates, as well as preserved antioxidant and antimicrobial activities. FTIR analysis and quantification results demonstrate the non-appearance of TTO on the microcapsule surfaces in any of the formulations, reaffirming that spray drying microencapsulation using natural biopolymers is a promising approach to overcome the limitations of TTO, such as high volatility and susceptibility to oxidation, and to improve stability and shelf life.

Our formulations showed uniform particle sizes. SEM images provided visual confirmation of our particle size data, revealing well-defined spherical structures with smooth surfaces, ensuring sustained preservation of EO functionality. The observed particle size distribution remained relatively narrow, despite some variation, which is characteristic of spray-dried particles.

The study’s results underscore the significant therapeutic potential of TTO and its microparticles for the treatment of ocular infections.

## Figures and Tables

**Figure 1 pharmaceutics-15-02489-f001:**
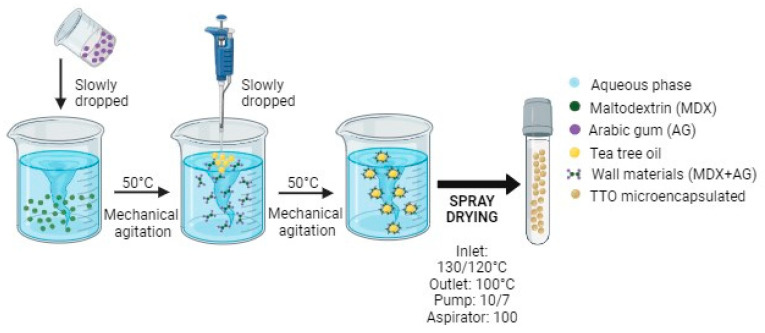
Schematic representation of the emulsion preparation and microencapsulation of Tea Tree Oil (TTO).

**Figure 2 pharmaceutics-15-02489-f002:**
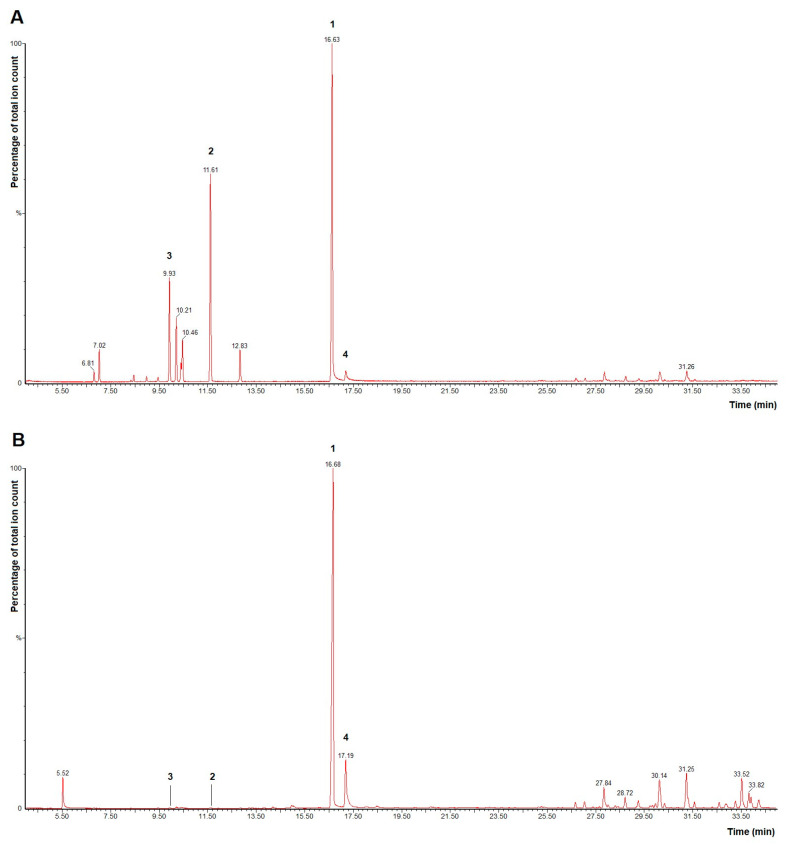
GC-MS profile of Tea Tree oil (TTO). (**A**) TTO prior to microencapsulation. (**B**) Encapsulated TTO. 1 = terpinen-4-ol, 2 = terpinolene, 3 = γ-terpinen, and 4 = α-terpineol.

**Figure 3 pharmaceutics-15-02489-f003:**
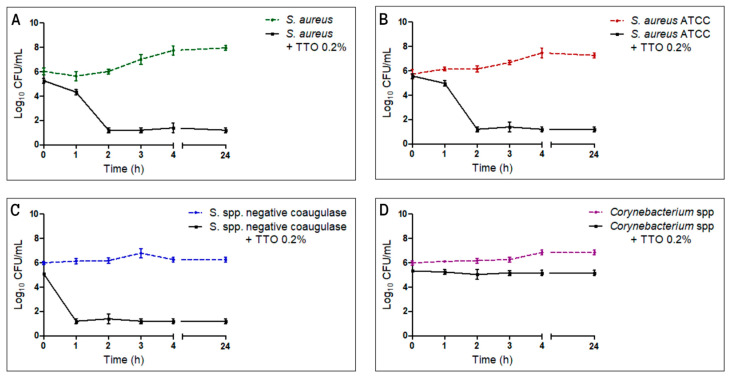
Time kill curves of (**A**) *S. aureus*; (**B**) *S. aureus* ATCC; (**C**) *Staphylococcus* spp. negative coagulase; and (**D**) *Corynebacterium* spp. Bacterial suspensions were incubated for 1, 2, 3, 4, and 24 h with TTO or without TTO (final concentration 0.2% *v*/*v*). Dash lines denote controls, whereas solid lines treatments in each panel. Samples were run in triplicates. Error bars represent the standard deviation of three independent experiments. TTO 0.2% *v*/*v*.

**Figure 4 pharmaceutics-15-02489-f004:**
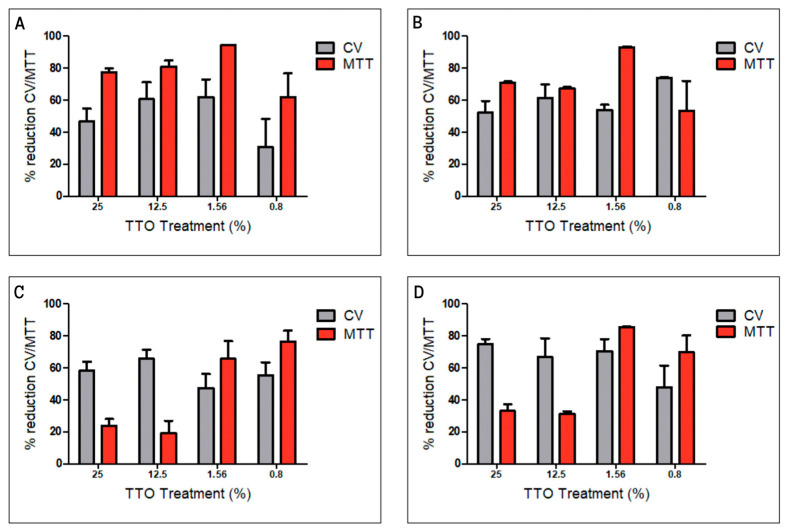
Activity of free TTO on biofilm formed by (**A**) *S. aureus*; (**B**) *S. aureus* ATCC; (**C**) *Staphylococcus* spp. negative coagulase; and (**D**) *Corynebacterium* spp. Each bar represents the percentage of biomass reduction evaluated with CV and the cell viability evaluated with MTT. Data represent the mean ± SD of six replicates of three independent experiments.

**Figure 5 pharmaceutics-15-02489-f005:**
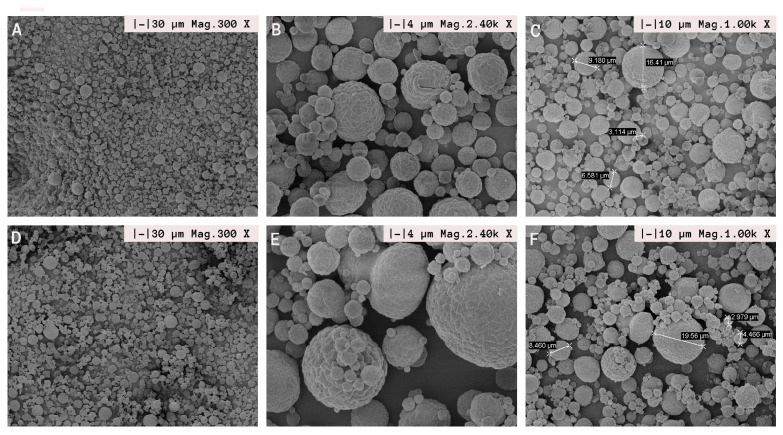
Photographs of SEM: (**A**) Formulation 1 at 300× magnification. (**B**) Formulation 1 at 2.40 k× magnification. (**C**) Formulation 1 at 1.00 k× magnification. (**D**) Formulation 2 at 300× magnification. (**E**) Formulation 2 at 2.40 k× magnification. (**F**) Formulation 2 at 1.00 k× magnification.

**Figure 6 pharmaceutics-15-02489-f006:**
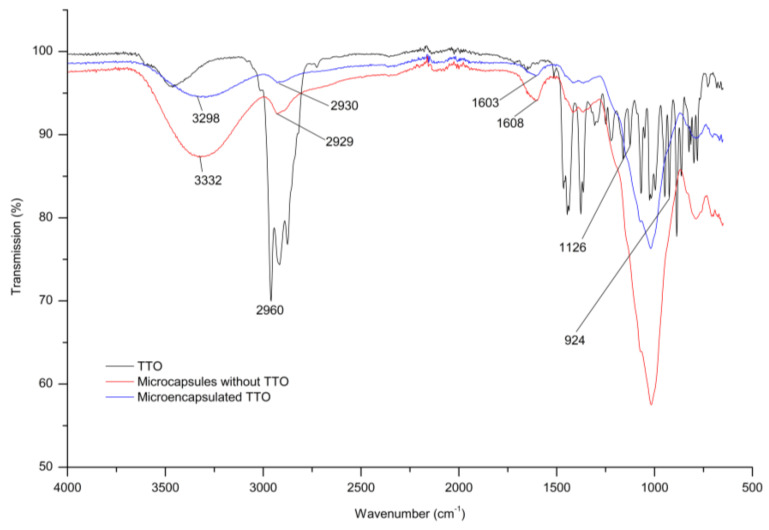
FTIR spectra of samples of Tea Tree Oil (TTO) and microcapsules with TTO and without TTO.

**Figure 7 pharmaceutics-15-02489-f007:**
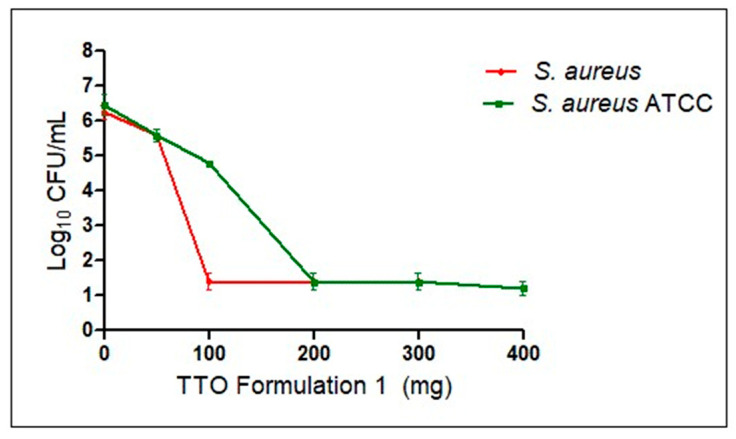
Viability assay depicting growth inhibition of *S. aureus* (red) and *S. aureus* ATCC (green) when exposed to different concentrations of Formulation 1 at 37° for 24 h. Data represent the mean ± standard deviation (SD) from three independent experiments (n = 3).

**Table 1 pharmaceutics-15-02489-t001:** Diffusion halo (mm) of tea tree oil inhibition (% *v*/*v*) against bacterial strains.

	Agar Diffusion Method (mm) ^1^				
Microorganism	TTO ^2^ % *v*/*v*				
	100	50	25	12.5	6.25
*S. aureus*	31 ± 2	25 ± 2	18 ± 2	14 ± 1	NH ^3^
*S. aureus ATCC*	29 ± 2	23 ± 1	15 ± 3	12 ± 2	10 ± 1
*Staphylococcus* spp. *negative coagulase*	31 ± 1	18 ± 3	11 ± 2	NH	NH
*Corynebacterium* spp.	21 ± 1	16 ± 1	14 ± 1	11 ± 2	8 ± 2

^1^ Millimeters of inhibition. ^2^ Tea tree oil. ^3^ Without the presence of halo.

**Table 2 pharmaceutics-15-02489-t002:** MIC and MBC (% *v*/*v*) values of tea tree oil against bacterial strains.

	MIC	MBC
Microorganism		
*S. aureus*	0.2% *v*/*v* (1.8 mg/mL)	0.4% *v*/*v* (3.5 mg/mL)
*S. aureus* ATCC	0.2% *v*/*v* (1.8 mg/mL)	0.4% *v*/*v* (3.5 mg/mL)
*Staphylococcus* spp. negative coagulase	0.4% *v*/*v* (3.5 mg/mL)	0.8% *v*/*v* (7.0 mg/mL)
*Corynebacterium* spp.	0.4% *v*/*v* (3.5 mg/mL)	- *

* Could not be measured.

**Table 3 pharmaceutics-15-02489-t003:** Particle size distribution Formulation 1 and 2.

	Particle Size (μm)			
	Average D_10_	Average D_50_	Average D_90_	Average Span
Formulation 1	6.17 ± 0.10	9.00 ± 0.20	12.90 ± 0.30	0.740 ± 0.002
Formulation 2	5.96 ± 0.09	8.56 ± 0.10	12.20 ± 0.30	0.720 ± 0.003

## Data Availability

Not applicable.

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
