# Peer review of "Enhancing the Functional Properties of Tea Tree Oil: In Vitro Antimicrobial Activity and Microencapsulation Strategy"

_pharmaceutics, 2023, doi:10.3390/pharmaceutics15102489_

Round 1
Reviewer 1 Report
The study is devoted to investigation of antimicrobial properties of tea tree oil (TTO) as a perspective biocidal agent. The micro-encapsulation via spray-drying method was applied to overcome TTO low water solubility and prone to oxidize when exposed to air.
The authors provided very detailed introduction (which might be slightly reduced) and performed a lot of antimicrobial tests (e.g., agar diffusion assay, MIC, MBC, anti-biofilm activity) of pure TTO and encapsulated oil. The characterisation of the obtained micro-capsules was also provided. Despite a number of the provided investigations, good structure of the manuscript and high scientific level of the work, some corrections must be done prior to publication.
Please, find below my corrections/suggestions/questions:
Lines 29-31: It is very strange to see that the MIC is measured in «%», but not in mg/ml or any other convenient units… Should be corrected.
Part 2.3.1.2, what is the initial concentration of TTO?
Lines 216-217: the sense of the phrase is not clear
Line 246: what «TTO.23» means?
Part 3.1, Fig. 2 should contain the information about TTO composition prior and after encapsulation.
Part 3.2.2
Lines 451-452: «The lowest dilution for MIC in the tested microorganisms was 0.2% of TTO, while for MBC it was 0.4%.» What was the initial TTO concentration? There is no information neither in experimental part nor on the results and discussion.
Again, why the concentration is given in «%» ?!
Line 617: what «drying.41» means ?
Part 3.6.2, there is no description of formulation 1 and formulation 2 neither in experimental part nor in discussion and results. What does every formulation contain ?
Parts 3.7 & 3.8 are incomplete, please, extend them.
The authors could also consider the use of FTIR to describe the properties of the obtained micro-capsules in both formulations.
The English level is good the minor revision is necessary.
Author Response
Dear reviewer,
All your comments and suggestions have been considered to improve the quality of the manuscript. We are grateful to you.
Every comment was answered as follows and in the final version of the manuscript, it is highlighted in yellow.
In the Comments and Suggestions for Authors section, the reviewer suggested that the introduction could be slightly shortened. This suggestion was taken into account, and it has been rewritten as follows for a clearer and more concise reading.
- Introduction
The pharmaceutical industry continually explores novel therapeutic alternatives for preventing and treating various diseases. Focusing on those that come from natural sources, an attractive option are essential oils (EOs) biosynthesized by plants. These natural products are mainly obtained by distillation applying different conventional and non-conventional extraction techniques [1]. This is feasible because EOs are a mixture of volatile substances of diverse chemical composition, mainly terpenes, phenylpropanoids and aromatic derivatives, which vary according to the species, and ambiental and anthropogenic factors.
EOs have been used for thousands of years in primary health care, demonstrating a broad spectrum of pharmacological activities [2,3]. Today, it is known that these medicinal properties are given because the components of EOs are capable of modulating numerous multiple signal transduction pathways, individually or in a synergistic manner.
Among the diverse pharmacological activities, antimicrobial properties of EOs have gained importance, particularly in light of the escalating challenge of microbial resistance [4]. In this scenario, EOs are surging as potential alternatives to antibiotics or as a complementary therapy alongside them.
One often overlooked healthcare issue pertains to periocular infections, which affect the area around the eyes, including the eyelids and the surrounding region. These infections can be caused by bacteria, viruses, or fungi. When left untreated or inadequately managed, they can progress and directly impact the eyes, leading to conditions like conjunctivitis, orbital cellulitis, chronic blepharitis, chorioretinitis, and endophthalmitis [5].
Both periocular and ocular infections are commonly treated with broad-spectrum antimicrobial drugs, often without proper pathogen identification through culture and susceptibility testing. This misuse can potentially promote antimicrobial resistance in ocular bacteria. In particular, bacterial biofilms are key contributors to resistance mechanisms, protecting the bacterial community [6].
Limited studies have been conducted on the use of essential oils (EOs) against bacteria from ocular infections, and even fewer studies on encapsulation systems that include EOs for this kind of pathologies are reported. However, using EOs in pharmaceutical applications presents challenges due to their physicochemical properties. Their unfavorable physicochemical properties, including hydrophobicity, low solubility in aqueous media, high volatility and oxygen mediated decomposition, and an undesired biological profile, like significant irritant action, restrict their applicability as therapeutic agents. Cosmetic and pharmaceutical formulations incorporating EOs have been developed to address these issues, yet stability problems persist. Encapsulation technologies, particularly microencapsulation via spray drying, offer a promising solution [7-10].
due to exposure to environmental factors like air, heat, light, and moisture substantially altering their composition during storage. Encapsulation technologies, particularly microencapsulation via spray drying, offer a promising solution [7-10].
Microencapsulation is an increasingly favored technique in the pharmaceutical industry due to its flexibility, cost-effectiveness, and suitability for heat-sensitive compounds [7-10]. It enables the production of ultrafine solid structures with high stability and encapsulation efficiency.
Incorporating antibacterial EOs into pharmaceutical formulations for periocular infections has the potential to enhance treatment efficacy, reduce the risk of antimicrobial resistance development, prevent its spread to ocular structures, and minimize the risk of serious complications that could endanger vision. Moreover, the complementary pharmacological properties of EOs, such as its antioxidant and anti- inflammatory effects, have the potential to enhance overall ocular health. These properties fortify the immune system and shield ocular tissues from oxidative damage caused by inflammatory processes.
Tea Tree Oil (TTO) presents a promising profile for an antimicrobial therapy. This oil is obtained mainly by steam distillation of the leaves of Melaleuca alternifolia (Cheel) Myrtaceae, a tree native to Australia [11,12]. ISO 4730:2017 standards establish that the main component of TTO is terpinen-4-ol, in a proportion not less than 40% [13]. Different studies on the subject have demonstrated the broad-spectrum antimicrobial activity of TTO, including antibacterial, antiprotozoal, antifungal, and antiviral activity [14-17].
With the passage of time, the gradual oxidation of components within TTO during the storage period can lead to a decrease in its antimicrobial effectiveness and potentially initiate undesired chemical reactions. Consequently, there is a growing demand for formulations that not only preserve the integrity of TTO but also enhance its inherent biological properties.
In view of these considerations, this study pursued a dual objective. Firstly, it assessed the antibacterial activity of a natural extract, TTO, against bacterial strains isolated from ocular infections. Secondly, it developed an encapsulation methodology using the spray drying technique to microencapsulate the selected EO. The resulting microcapsules underwent various analyses, encompassing the evaluation of their physical and morphological characteristics, in vitro drug release profiles, and investigations through scanning electron microscopy.
Therefore, further research and exploration of TTO as a promising therapeutic option for eye infections is essential.
1- Lines 29-31: It is very strange to see that the MIC is measured in «%», but not in mg/ml or any other convenient units… Should be corrected.
Author's response: Thank you. The unit of concentration in which the MIC was determined has been corrected and added in the main text. We have converted the unit from %v/v to mg/mL.
2- Part 2.3.1.2, what is the initial concentration of TTO?
Author's response: -Thank you. The concentration of TTO is 1 mL per milliliter (1mL/mL). The phrase "The concentration of TTO is 1 mL per milliliter (1mL/mL)" has been added to the Materials and Methods section.
3- Lines 216-217: the sense of the phrase is not clear.
Author's response: -Thank you. The phrase “A method by which the in vitro activity of different concentrations of TTO is determined against a microorganism during a determined period of time” changed to “Time kill curves illustrate bacterial elimination over time as a function of the concentration of TTO”.
4- Line 246: what «TTO.23» means?
Author's response: -Thank you. The number 23 that appears in that line corresponds to a bibliographic reference. It has already been included as required by the journal's author guidelines. TTO.23 changed to TTO [23].
5- Part 3.1, Fig. 2 should contain the information about TTO composition prior and after encapsulation.
Author's response: -Thank you for your suggestions. The following text and Figure 2 A and B, related to the TTO composition before and after encapsulation, are now displayed in the mentioned section.
3.1. GC-MS analysis of TTO and encapsulated TTO
Composition of the commercial sample of TTO used was compared with that reported in literature. The percentage of the chemical marker stated by ISO 4730:2017, terpinen-4-ol, was determined as 44,8%, as well as other highlighted components (Figure 2A). It is important to point out that o-cymene, a decomposition indicator, was detected at a low percentage in this analysis.
On the other hand, after microencapsulation, the composition of TTO inside the capsule (formulation 1) was also analyzed, and a similar fingerprint was observed, where terpinen-4-ol was the main component (52,1%) (Figure 2B).
Figure 2. Composition of Tea Tree oil (TTO) analyzed by GC-MS. (A) TTO prior to microencapsulation. (B) Encapsulated TTO. 1= terpinen-4-ol, 2= terpinolene, 3= γ-terpinen, 4= α-terpineol
6- Part 3.2.2
6.1- Lines 451-452: «The lowest dilution for MIC in the tested microorganisms was 0.2% of TTO, while for MBC it was 0.4%.» What was the initial TTO concentration? There is no information neither in experimental part nor on the results and discussion.
Author's response: -Thank you. The concentration of TTO is 1 mL per milliliter (1 mL/mL), and this information was included in the materials and methods section. The unit of concentration in which the MIC was determined has been corrected.
6.2- Again, why the concentration is given in «%»?!
Author's response: -Thank you. The unit of concentration in which the MIC was determined has been corrected. «%» was modified to mg/mL.
7- Line 617: what «drying.41» means?
Author's response: -Thank you. The number 41 that appears in that line corresponds to a bibliographic reference. It has already been included as required by the journal's author guidelines. drying.41 was modified to drying [41].
8- Part 3.6.2, there is no description of formulation 1 and formulation 2 neither in experimental part nor in discussion and results. What does every formulation contain?
Author's response: -Thank you. The following parameters were fixed. For emulsion 1: pump 10, aspirator 100; Q-flow, 600 L/h; inlet temperature, 130°C; outlet temperature, 100°C. The same parameters were used for emulsion 2 except for the temperature which was 120°C and the pump 7 respectively. The ME was about 90-95% for both microcapsule preparations. The ME of TTO is the ratio between the total TTO within the microcapsules and the surface TTO. The ER, calculated as the ratio between the total TTO and the theoretical TTO, was about 40%.
9- Parts 3.7 & 3.8 are incomplete, please, extend them.
Author's response: -Thank you. We have modified these points based on your suggestions.
10- The authors could also consider the use of FTIR to describe the properties of the obtained micro-capsules in both formulations.
Author's response: -Thank you. The method was utilized and detailed in the revised manuscript, appearing in both the Materials and Methods (2.5.6.) and Results and Discussion (3.6.2.) sections. This approach substantiated the effective microencapsulation process, as there was no observation of tea tree oil on the microcapsule surfaces; however, oil quantification was achieved subsequent to its release through diverse techniques.
2.5.6. Fourier-Transform Infrared Spectroscopy (FTIR)
The Fourier-Transform Infrared Spectroscopy (FTIR) analysis of TTO in its pure form, as well as the mixture of wall material and structures after spray drying, was performed on a droplet of each sample. The equipment used was a CARY 630 FTIR (Agilent Technology, Santa Clara, CA) covering a range of 500 to 4000 cm-1 with a resolution of 3 cm-1. Sixteen scans were performed for each sample analyzed.
3.6.2. Particle size and morphological characterization of the microcapsules of TTO
The FTIR spectra of TTO and the microcapsules with and without TTO are presented in Figure 6.
The spectrum of TTO revealed a stretching vibration peak corresponding to the C-H bond at 2960 cm-1. Additionally, it exhibited numerous peaks within the wavelength range of 1550 cm-1 to 600 cm-1. The peak at 1126 cm-1 was attributed to the stretching vibration absorption peak of the C-O bond in the tertiary alcohol of terpenes and terpineol, respectively. The peak at 924 cm-1 corresponded to the bending vibration absorption peak of an unsaturated double bond.
For the microcapsules without TTO, characteristic hydroxyl peaks (O-H stretching) were observed around 3332 cm-1. Peaks corresponding to the C-H stretching from the carboxylic group appeared around 2929 cm-1. Peaks corresponding to amine or carbonyl groups appeared at 1608 cm-1. For the microcapsules with TTO, characteristic hydroxyl peaks (O-H stretching) were observed around 3298 cm-1. Peaks corresponding to the C-H stretching from the carboxylic group appeared around 2930 cm-1. Peaks corresponding to amine or carbonyl groups appeared at 1603 cm-1. All of these peaks are characteristic of the wall material used in the formulation design. These bands representing coating materials were evident in the microcapsule spectra. This implies that these carbohydrates maintained their structure during the drying process.
The spectra of the microcapsules with and without TTO confirmed the successful microencapsulation through spray drying, with no TTO present on the surfaces of the microcapsules [48].
Comments on the Quality of English Language. The English level is good the minor revision is necessary.
We are very grateful for your feedback. The manuscript's English language review has been carried out.

Reviewer 2 Report
While I would like to thank the authors for supplying the manuscript "Enhancing the functional properties of tea tree oil: in vitro antimicrobial activity and microencapsulation strategy". The authors should be commended for attempting to perform wide ranging characterisation both analytically and microbiologically. However I feel the discussion and analysis of the experimental work is not currently up to the required standard.
The lack of in-depth discussion on the relevance of the experimental results can hinder the practical application of the research. Without a clear understanding of how these findings relate to real-world scenarios or potential consumer benefits, the research may remain in the realm of academic curiosity rather than practical use.
Figure 1 is overly simplistic and use of similar colours for makes it difficult to understand the differences between the steps. I would also label the temperature in each step as the schematic should be self explanatory and not require reading the description also( This suggestion is minor in nature and not essential if access to to the original vector images is not available. The step marked excipient could also be changed to well / wall material as this is the term used in the description. Also harmonise the use of well / wall material as both are used in section 2.5.1
Section 2.5.2 - It was stated a two fluid nozzle was used but the second fluid not mentioned.
When referring to the supplied equations different annotations were used . Squared brackets in Line 317, and on Line 340 it was referred to as (eq. 5.)
Table 1 - Is a value of 0 +/- 1 technically possible
Figure 3: The resolution of the supplied graph is not suitable, I would also consider using 4 separate graphs for each ocular infection to show the difference between treated and untreated.
Results are not provided for sections 3.7 and 3.8
Placeholder text from the Pharmaceutics template remains in Section 3.8 and Reference 47
The conclusion does not use any of the results from the study.
Only minor editing of the English language is required in the current version
Author Response
REVIEWER N°2
Dear reviewer,
All your comments and suggestions have been considered to improve the quality of the manuscript. We are grateful to you.
Every comment was answered as follows and in the final version of the manuscript, it is highlighted in yellow.
1- Figure 1 is overly simplistic and use of similar colours for makes it difficult to understand the differences between the steps. I would also label the temperature in each step as the schematic should be self explanatory and not require reading the description also( This suggestion is minor in nature and not essential if access to to the original vector images is not available. The step marked excipient could also be changed to well / wall material as this is the term used in the description. Also harmonise the use of well / wall material as both are used in section 2.5.1
Author's response: -Thank you. All the suggestions were considered in the new Figure 1, which is included in the final version of the manuscript.
Figure 1. Schematic representation of the emulsion preparation and microencapsulation of Tea Tree Oil (TTO).
2- Section 2.5.2 - It was stated a two fluid nozzle was used but the second fluid not mentioned.
Author's response: -Thank you. The two-fluid nozzle operates on the basic principle of using high-speed air (first fluid) to break down the liquid (emulsions for microencapsulation, second fluid), resulting in smaller liquid particles and higher flow rates.
3- When referring to the supplied equations different annotations were used. Squared brackets in Line 317, and on Line 340 it was referred to as (eq. 5.)
Author's response: -Thank you. The square brackets in line 317 were replaced with parentheses, thereby standardizing the way it is expressed in the manuscript. [Equation (3)] was modified to (eq. 3).
4- Table 1 - Is a value of 0 +/- 1 technically possible
Author's response: -Thank you. The changes have been made to Table 1 according to the reviewer's suggestions.
5- Figure 3: The resolution of the supplied graph is not suitable, I would also consider using 4 separate graphs for each ocular infection to show the difference between treated and untreated.
Author's response: -Thank you. We have considered your suggestion, and we have designed Figure 3 with 4 separate graphs to illustrate the noted differences.
Figure 3. Time kill curves of (A) S. aureus; (B) S. aureus ATCC; (C) Staphylococcus spp. negative coagulase; and (D) Corynebacterium spp. Bacterial suspensions were incubated for 1, 2, 3, 4, and 24 hours with TTO or without TTO (final concentration 0.2%v/v). Dash lines denote controls, whereas solid lines treatments in each panel. Samples were run in triplicates. Error bars represent the standard deviation of three independent experiments. TTO 0.2%v/v
6- Results are not provided for sections 3.7 and 3.8
Author's response: -Thank you. We have modified these points based on your suggestions.
7- Placeholder text from the Pharmaceutics template remains in Section 3.8 and Reference 47
Author's response: -Thank you. The highlighted placeholder text has been removed from Section 3.8 and Reference 47.
8- The conclusion does not use any of the results from the study.
Author's response: -Thank you. Your suggestion was taken into account, and the conclusion has been rewritten as follows, incorporating the results of the studies conducted.
TTO has demonstrated efficacy against bacterial strains isolated from ocular infections, including Corynebacterium spp., Staphylococcus spp. negative for coagulase and Staphylococcus aureus, as well as a reference strain of Staphylococcus aureus (ATCC 25923). TTO exhibited a substantial reduction in biofilm biomass, ranging from 30% to 70%.
The microencapsulation technique used in the study successfully prepared TTO-containing formulations with high encapsulation yields, microencapsulation efficiency, and embedding rates, as well as preserved antioxidant and antimicrobial activities. FTIR analysis and quantification results demonstrate the non-appearance of TTO on the microcapsule surfaces in any of the formulations, reaffirming that spray drying microencapsulation using natural biopolymers is a promising approach to overcome the limitations of TTO, such as high volatility and susceptibility to oxidation, and to improve stability and shelf life.
Our formulations showed uniform particle sizes. SEM images provided visual confirmation of our particle size data, revealing well-defined spherical structures with smooth surfaces, ensuring sustained preservation of EO functionality. The observed particle size distribution remained relatively narrow, despite some variation, which is characteristic of spray-dried particles.
The study's results underscore the significant therapeutic potential of TTO and its microparticles for the treatment of ocular infections.
Reviewer 3 Report
In this article, the authors reported microcapsules containing TTO for anti-microbial. The TTO emulsion is prepared by first high-power homogenizing the Maltodextrin and Arabic gum, then the emulsifying the TTO in the solution of Maltodextrin and Arabic gum, the solution is then spray drying and acquire the TTO microcapsule. I suggest this article to be published, but before publishing, here are some issues need to be addressed.
1. The authors should calculate or measure the load efficiency of the TTO microcapsule.
2. Since the authors mentioned using the DLS for measuring the size of the microcapsules, how about the size distribution and PDI of the microcapsules. These data are needed to be discussed.
3. From Figure 5, the size of the TTO microcapsules is not uniform, the authors need to discuss the factors influence the size distribution of the microcapsules. And how to prove the microsphere are microcapsules rather than particles?
4. The thickness of the microcapsules are needed to discuss and calculate, I suggest the authors to supplement the figures which could both investigate and calculate the thickness of the microcapsules.
5. Some articles about microcapsules are suggested to cite and discuss, such as Chinese Chemical Letters, 2020, 31, 249; Small 2020, 16, 2002716.
Good.
Author Response
Dear reviewer,
All your comments and suggestions have been considered to improve the quality of the manuscript. We are grateful to you.
Every comment was answered as follows and in the final version of the manuscript, it is highlighted in yellow.
- The authors should calculate or measure the load efficiency of the TTO microcapsule.
Author's response: -Thank you. We have modified this point based on your suggestions.
- Since the authors mentioned using the DLS for measuring the size of the microcapsules, how about the size distribution and PDI of the microcapsules. These data are needed to be discussed.
Author's response: -Thank you. We have modified this point based on your suggestions.
The samples were atomized with a hot air stream in a drying chamber, thus making it possible to obtain solid microparticles where the EO was trapped within a film of encapsulating material.
Particle size is important as it can affect the microencapsulation efficiency, as well as the interaction with other fluids. The particle size of a microcapsule ranges from 1 to 1000 µm depending on the manufacturing method and its specific purpose. Particle size is considered a critical aspect for encapsulating substances and is an important factor in the controlled release of bioactive agents [40,41].
All the parameters described in the methods were analyzed, resulting in an average diameter (d50) of 6 µm for both formulations. The most relevant value was that of d90, which characterizes the majority of the particle population. The D90 values obtained were 12.90 ± 0.30 µm and 12.20 ± 0.30 µm for formulations 1 and 2, respectively, reflecting a uniform particle size, as expected for them to be considered microparticles. Furthermore, the calculated span value demonstrates a narrow particle size distribution for both formulations across all replicates.
A slightly broader particle distribution was recorded, ranging between 8 and 15 μm, when microencapsulating the EO of Lippia sidoides Cham. (Verbenaceae) [42] and the microencapsulation of Origanum vulgare L. essential oil (7–18 μm) using a combination of MX, AG, and modified starch [43].
In our study, the smallest particle size was recorded with formulation 2 (12.20 μm). However, no significant differences were found in particle size compared to formulation 1 when variations in equipment parameters were made prior to spray drying. For these reasons, both formulations could be used as potential strategies for microencapsulating TTO.
With the purpose of ensuring proper administration of a microencapsulated active ingredient, it is advisable for the microparticle size to be sufficiently small [44].
- From Figure 5, the size of the TTO microcapsules is not uniform, the authors need to discuss the factors influence the size distribution of the microcapsules. And how to prove the microsphere are microcapsules rather than particles?
Author's response: -Thank you. We have modified this point based on your suggestions.
- The thickness of the microcapsules are needed to discuss and calculate, I suggest the authors to supplement the figures which could both investigate and calculate the thickness of the microcapsules.
Author's response: -Thank you. We have modified this point based on your suggestions.
The obtained SEM images play a crucial role, not only in verifying the acquired particle size data but also in assessing the morphological features that elucidate the functional attributes of the particles, such as their capacity to retain and safeguard the bioactive ingredient [45].
The images captured revealed distinct individual spherical structures with smooth surfaces, devoid of any cracks or fissures that might allow the release of TTO. This ensures the sustained preservation of its functionality over time.
The SEM images are consistent with the obtained span value. While a range of sizes is evident, which is a characteristic feature of particles produced by spray drying due to factors like the material used, formulation, and atomization process [41], the observed size range remains relatively narrow. In a study involving the microencapsulation of lavender essential oil, it was observed that smaller particles fused with larger ones, as evident in images B and D [46,47].
- Some articles about microcapsules are suggested to cite and discuss, such as Chinese Chemical Letters, 2020, 31, 249; Small 2020, 16, 2002716.
Author's response: -Thank you. We have taken into account the articles suggested in the discussion.
Comments on the Quality of English Language Good.
We are very grateful for your feedback. The manuscript's English language review has been carried out.

Round 2
Reviewer 1 Report
Dear authors,
First of all I'd like to thank the authors for the provided revision of the manuscript. Nevertheless your work still has some flaws that should be corrected prior to its publication in Pharmaceutics.
1. The Figure 2 should be considerably ameliorated: please, provide the names of the axis, as well as the corresponding units. I presume that the x-axis is the time. In this case, please, synchronize the time scale (including the increment values) of the figures A and B. Moreover, the sense of the figure is not completely correct: the figure presents the chromatograms but not the composition.
2. Please provide a separate table containing the information on formulation 1 and 2 in 2.5.2.
3. Parts 2.3.1.2 and 3.2.2, please provide TTO concentrations in moles/ml or in mg/ml.
4. Part 3.7 is too short, the extended discussion including the comparison of the obtained results with the literature data (the similar type of encapsulated antimicrobial composites) should be provided.
5. The quality of all provided figures in the manuscript (with exception of Fig.5) is rather mediocre. Please, save the figures in higher dpi resolution. It would be nice if all figures would follow the same style.
A little revision of the grammar is required.
Author Response
Dear reviewer,
We sincerely appreciate the recommendations and suggestions you have provided during the review process. Your contributions have been extremely valuable and are clearly evident in the final manuscript's outcome. We have worked diligently to incorporate your comments and enhance the quality of the work.
Your dedication to the review has significantly contributed to the manuscript's quality and robustness, and we are very grateful for it. We hope that with these improvements, the work can have an even more positive impact on the research field.
Once again, we thank you for your time and effort in this process.
Dear reviewer,
All your comments and suggestions have been considered to improve the quality of the manuscript. We are grateful to you.
Every comment was answered as follows and in the final version of the manuscript, it is highlighted in yellow.
- The Figure 2 should be considerably ameliorated: please, provide the names of the axis, as well as the corresponding units. I presume that the x-axis is the time. In this case, please, synchronize the time scale (including the increment values) of the figures A and B. Moreover, the sense of the figure is not completely correct: the figure presents the chromatograms but not the composition.
Author's response: We really appreciate the comments made to improve the interpretation of Figure 2. Labels were added to axes and the time scale was synchronized. To give an appropriate sense to the figure, the legend was modified as well.
- Please provide a separate table containing the information on formulation 1 and 2 in 2.5.2.
Author's response: In the materials and methods section (2.5.2.), the difference between the two formulations presented was described as follows: "For emulsion 1: pump 10, aspirator 100; Q-flow, 600 L/h; inlet temperature, 130°C; outlet temperature, 100°C. The same parameters were used for emulsion 2 except for the inlet temperature, which was 120°C, and the pump, which was set to 7."
We would like to highlight that there were no differences observed in the composition with respect to the wall components. However, significant variations were noted when adjusting the inlet temperature, which was changed from 130°C to 120°C. This modification was made because higher temperatures led to noticeable alterations in the oil composition. We believe that a table with this information is not necessary.
- Parts 2.3.1.2 and 3.2.2, please provide TTO concentrations in moles/ml or in mg/ml.
Author's response: TTO concentrations have been provided in mg/mL in the specified sections.
- Part 3.7 is too short, the extended discussion including the comparison of the obtained results with the literature data (the similar type of encapsulated antimicrobial composites) should be provided.
Author's response:
Various concentrations of Formulation 1 were meticulously employed to assess its antimicrobial effectiveness over a period spanning 24 h. The chosen test strains for this evaluation included a reference strain of S. aureus ATCC and a clinical isolate of S. aureus. The graphical representation in Figure 7 eloquently illustrates the profound impact of Formulation 1 on cellular viability, showcased over the course of the incubation period. The data presents a strikingly evident bactericidal profile for both of the aforementioned bacterial strains, underscoring a significant reduction in bacterial viability, surpassing an impressive 99.99% reduction concerning the control group. These results serve as compelling evidence of the successful and efficacious release of the active compound, TTO, within Formulation 1, solidifying its potential as a potent antimicrobial agent.
- The quality of all provided figures in the manuscript (with exception of Fig.5) is rather mediocre. Please, save the figures in higher dpi resolution. It would be nice if all figures would follow the same style.
Author's response: We appreciate your feedback. We have recreated the figures with higher resolution while maintaining the same style.
Comments on the Quality of English Language
A little revision of the grammar is required.
Author's response: We are very grateful. The grammar of the manuscript has been reviewed.

Reviewer 2 Report
I would like to thank the authors for their timely resubmission of the revised manuscript. There is an immediate visible improvement in the quality of the manuscript and I hope the authors feel that the peer review process has been worthwhile and has resulted in an improved manuscript.
There are some very minor comments that can be addressed prior to final submission/publication as highlighted below.
Line 82- Missing section of capital letter for 'Due'
Line 314-319 Alignment of equations is needed
Author Response
Dear reviewer,
We sincerely appreciate the recommendations and suggestions you have provided during the review process. Your contributions have been extremely valuable and are clearly evident in the final manuscript's outcome. We have worked diligently to incorporate your comments and enhance the quality of the work.
Your dedication to the review has significantly contributed to the manuscript's quality and robustness, and we are very grateful for it. We hope that with these improvements, the work can have an even more positive impact on the research field.
Once again, we thank you for your time and effort in this process.
There are some very minor comments that can be addressed prior to final submission/publication as highlighted below.
Author's response: We appreciate your comments, which will be included in the final manuscript attached.
Line 82- Missing section of capital letter for 'Due'
Author's response: The sentence has been rephrased. We appreciate your clarification.
Line 314-319 Alignment of equations is needed
Author's response: The equations have been aligned. We appreciate your feedback.

Reviewer 3 Report
I agree this article to be published now.
Good.
Author Response
Dear reviewer,
We sincerely appreciate the recommendations and suggestions you have provided during the review process. Your contributions have been extremely valuable and are clearly evident in the final manuscript's outcome. We have worked diligently to incorporate your comments and enhance the quality of the work.
Your dedication to the review has significantly contributed to the manuscript's quality and robustness, and we are very grateful for it. We hope that with these improvements, the work can have an even more positive impact on the research field.
Once again, we thank you for your time and effort in this process.
